# HYBRID AND NON-UNIFORM QUANTIZATION METHODS USING RETRO SYNTHESIS DATA FOR EFFICIENT INFERENCE

## ABSTRACT

Existing quantization aware training methods attempt to compensate for the quantization loss by leveraging on training data, like most of the post-training quantization methods, and are also time consuming. Both these methods are not effective for privacy constraint applications as they are tightly coupled with training data. In contrast, this paper proposes a data-independent post-training quantization scheme that eliminates the need for training data. This is achieved by generating a faux dataset, hereafter referred to as *'Retro-Synthesis Data'*, from the FP32 model layer statistics and further using it for quantization. This approach outperformed state-of-the-art methods including, but not limited to, ZeroQ and DFQ on models with and without Batch-Normalization layers for 8, 6, and 4 bit precisions on ImageNet and CIFAR-10 datasets. We also introduced two futuristic variants of post-training quantization methods namely *'Hybrid Quantization'* and *'Non-Uniform Quantization'*. The Hybrid Quantization scheme determines the sensitivity of each layer for per-tensor & per-channel quantization, and thereby generates hybrid quantized models that are **'10 to 20%'** efficient in inference time while achieving the same or better accuracy compared to per-channel quantization. Also, this method outperformed FP32 accuracy when applied for ResNet-18, and ResNet-50 models on the ImageNet dataset. In the proposed Non-Uniform Quantization scheme, the weights are grouped into different clusters and these clusters are assigned with a varied number of quantization steps depending on the number of weights and their ranges in the respective cluster. This method resulted in **'1%'** accuracy improvement against state-of-the-art methods on the ImageNet dataset.

## 1 INTRODUCTION

Quantization is a widely used and necessary approach to convert heavy Deep Neural Network (DNN) models in Floating Point (FP32) format to a light-weight lower precision format, compatible with edge device inference. The introduction of lower precision computing hardware like *Qualcomm Hexagon DSP* (Codrescu, 2015) resulted in various quantization methods (Morgan et al., 1991; Rastegari et al., 2016; Wu et al., 2016; Zhou et al., 2017; Li et al., 2019; Dong et al., 2019; Krishnamoorthi, 2018) compatible for edge devices. Quantizing a FP32 DNN to INT8 or lower precision results in model size reduction by at least $4X$ based on the precision opted for. Also, since the computations happen in lower precision, it implicitly results in faster inference time and lesser power consumption. The above benefits with quantization come with a caveat of accuracy loss, due to noise introduced in the model's weights and activations.

In order to reduce this accuracy loss, quantization aware fine-tuning methods are introduced (Zhu et al., 2016; Zhang et al., 2018; Choukroun et al., 2019; Jacob et al., 2018; Baskin et al., 2019; Courbariaux et al., 2015), wherein the FP32 model is trained along with quantizers and quantized weights. The major disadvantages of these methods are, they are computationally intensive and time-consuming since they involve the whole training process. To address this, various post-training quantization methods (Morgan et al., 1991; Wu et al., 2016; Li et al., 2019; Banner et al., 2019) are developed that resulted in trivial to heavy accuracy loss when evaluated on different DNNs. Also, to determine the quantized model's weight and activation ranges most of these methods require access to training data, which may not be always available in case of applications with security and privacy

constraints which involve card details, health records, and personal images. Contemporary research in post-training quantization (Nagel et al., 2019; Cai et al., 2020) eliminated the need for training data for quantization by estimating the quantization parameters from the Batch-Normalization (BN) layer statistics of the FP32 model but fail to produce better accuracy when BN layers are not present in the model.

To address the above mentioned shortcomings, this paper proposes a data-independent post-training quantization method that estimates the quantization ranges by leveraging on 'retro-synthesis' data generated from the original FP32 model. This method resulted in better accuracy as compared to both data-independent and data-dependent state-of-the-art quantization methods on models ResNet18, ResNet50 (He et al., 2016), MobileNetV2 (Sandler et al., 2018), AlexNet (Krizhevsky et al., 2012) and ISONet (Qi et al., 2020) on ImageNet dataset (Deng et al., 2009). It also outperformed state-of-the-art methods even for lower precision such as 6 and 4 bit on ImageNet and CIFAR-10 datasets. The 'retro-synthesis' data generation takes only *10 to 12 sec* of time to generate the entire dataset which is a minimal overhead as compared to the benefit of data independence it provides. Additionally, this paper introduces two variants of post-training quantization methods namely *'Hybrid Quantization'* and *'Non-Uniform Quantization'*.

## 2 PRIOR ART

### 2.1 QUANTIZATION AWARE TRAINING BASED METHODS

An efficient integer only arithmetic inference method for commonly available integer only hardware is proposed in Jacob et al. (2018), wherein a training procedure is employed which preserves the accuracy of the model even after quantization. The work in Zhang et al. (2018) trained a quantized bit compatible DNN and associated quantizers for both weights and activations instead of relying on handcrafted quantization schemes for better accuracy. A 'Trained Ternary Quantization' approach is proposed in Zhu et al. (2016) wherein the model is trained to be capable of reducing the weights to 2-bit precision which achieved model size reduction by 16x without much accuracy loss. Inspired by other methods Baskin et al. (2019) proposes a 'Compression Aware Training' scheme that trains a model to learn compression of feature maps in a better possible way during inference. Similarly, in binary connect method (Courbariaux et al., 2015) the network is trained with binary weights during forward and backward passes that act as a regularizer. Since these methods majorly adopt training the networks with quantized weights and quantizers, the downside of these methods is not only that they are time-consuming but also they demand training data which is not always accessible.

### 2.2 POST TRAINING QUANTIZATION BASED METHODS

Several post-training quantization methods are proposed to replace time-consuming quantization aware training based methods. The method in Choukroun et al. (2019) avoids full network training, by formalizing the linear quantization as 'Minimum Mean Squared Error' and achieves better accuracy without retraining the model. 'ACIQ' method (Banner et al., 2019) achieved accuracy close to FP32 models by estimating an analytical clipping range of activations in the DNN. However, to compensate for the accuracy loss, this method relies on a run-time per-channel quantization scheme for activations which is inefficient and not hardware friendly. In similar lines, the OCS method (Zhao et al., 2019) proposes to eliminate the outliers for better accuracy with minimal overhead. Though these methods considerably reduce the time taken for quantization, they are unfortunately tightly coupled with training data for quantization. Hence they are not suitable for applications wherein access to training data is restricted. The contemporary research on data free post-training quantization methods was successful in eliminating the need for accessing training data. By adopting a per-tensor quantization approach, the DFQ method (Nagel et al., 2019) achieved accuracy similar to the per-channel quantization approach through cross layer equalization and bias correction. It successfully eliminated the huge weight range variations across the channels in a layer by scaling the weights for cross channels. In contrast ZeroQ (Cai et al., 2020) proposed a quantization method that eliminated the need for training data, by generating distilled data with the help of the Batch-Normalization layer statistics of the FP32 model and using the same for determining the activation ranges for quantization and achieved state-of-the-art accuracy. However, these methods tend to observe accuracy degradation when there are no Batch-Normalization layers present in the FP32 model.

To address the above shortcomings the main contributions in this paper are as follows:

- A data-independent post-training quantization method by generating the *'Retro Synthesis'* data, for estimating the activation ranges for quantization, without depending on the Batch-Normalization layer statistics of the FP32 model.
- Introduced a *'Hybrid Quantization'* method, a combination of Per-Tensor and Per-Channel schemes, that achieves state-of-the-art accuracy with lesser inference time as compared to fully per-channel quantization schemes.
- Recommended a *'Non-Uniform Quantization'* method, wherein the weights in each layer are clustered and then allocated with a varied number of bins to each cluster, that achieved **'1%'** better accuracy against state-of-the-art methods on ImageNet dataset.

## 3 METHODOLOGY

This section discusses the proposed data-independent post-training quantization methods namely (a) Quantization using retro-synthesis data, (b) Hybrid Quantization, and (c) Non-Uniform Quantization.

### 3.1 QUANTIZATION USING RETRO SYNTHESIS DATA

In general, post-training quantization schemes mainly consist of two parts - (i) quantizing the weights that are static in a given trained FP32 model and (ii) determining the activation ranges for layers like ReLU, Tanh, Sigmoid that vary dynamically for different input data. In this paper, asymmetric uniform quantization is used for weights whereas the proposed 'retro-synthesis' data is used to determine the activation ranges. It should be noted that we purposefully chose to resort to simple asymmetric uniform quantization to quantize the weights and also have not employed any advanced techniques such as outlier elimination of weight clipping for the reduction of quantization loss. This is in the interest of demonstrating the effectiveness of 'retro-synthesis' data in accurately determining the quantization ranges of activation outputs. However, in the other two proposed methods (b), and (c) we propose two newly developed weight quantization methods respectively for efficient inference with improved accuracy.

### 3.1.1 RETRO-SYNTHESIS DATA GENERATION

Aiming for a data-independent quantization method, it is challenging to estimate activation ranges without having access to the training data. An alternative is to use "random data" having Gaussian distribution with 'zero mean' and 'unit variance' which results in inaccurate estimation of activation ranges thereby resulting in poor accuracy. The accuracy degrades rapidly when quantized for lower precisions such as 6, 4, and 2 bit. Recently ZeroQ (Cai et al., 2020) proposed a quantization method using distilled data and showed significant improvement, with no results are showcasing the generation of distilled data for the models without Batch-Normalization layers and their corresponding accuracy results.

In contrast, inspired by ZeroQ (Cai et al., 2020) we put forward a modified version of the data generation approach by relying on the fact that, DNNs which are trained to discriminate between different image classes embeds relevant information about the images. Hence, by considering the class loss for a particular image class and traversing through the FP32 model backward, it is possible to generate image data with similar statistics of the respective class. Therefore, the proposed *"retro-synthesis"* data generation is based on the property of the trained DNN model, where the image data that maximizes the class score is generated, by incorporating the notion of the class features captured by the model. Like this, we generate a set of images corresponding to each class using which the model is trained. Since the data is generated from the original model itself we named the data as *"retro-synthesis"* data. It should be observed that this method has no dependence on the presence of Batch-Normalization layers in the FP32 model, thus overcoming the downside of ZeroQ. It is also evaluated that, for the models with Batch-Normalization layers, incorporating the proposed *"class-loss"* functionality to the distilled data generation algorithm as in ZeroQ results in improved accuracy. The proposed *"retro-synthesis"* data generation method is detailed in Algorithm 1. Given, a fully trained FP32 model and a class of interest, our aim is to empirically generate an image that

is representative of the class in terms of the model class score. More formally, let $P(C)$ be the soft-max of the class $C$, computed by the final layer of the model for an image $I$. Thus, the aim is, to generate an image such that, this image when passed to the model will give the highest softmax value for class $C$.

---

**Algorithm 1** Retro synthesis data generation

---

**Input:** Pre-determined FP32 model (M), Target class (C).
**Output:** A set of retro-synthesis data corresponding to Target class (C).

1. **Init:** $I \leftarrow$ random_gaussian(batch-size, input shape)

2. **Init:** $Target \leftarrow$rand(No. of classes) $\ni argmax(Target) = C$

3. **Init:** $\mu_0 = 0, \sigma_0 = 1$

4. Get $(\mu_i, \sigma_i)$ from batch norm layers of M (if present), $i \in 0, 1, \ldots, n$ where $n \rightarrow$ No.of batch norm layers

5. **for** $j = 0, 1, \ldots,$No. of Epochs

   (a) Forward propagate $I$ and gather intermediate activation statistics

   (b) $Output = M(I)$

   (c) $Loss_{BN}$=0

   (d) **for** $k = 0, 1, \ldots, n$

      i. **Get** $(\mu_k, \sigma_k)$
      ii. $Loss_{BN} \leftarrow Loss_{BN}$+$L((\mu_k, \sigma_k), (\mu_k^{BN}, \sigma_k^{BN}))$

   (e) **Calculate** $(\mu_0', \sigma_0')$ of $I$

   (f) $Loss_G \leftarrow L((\mu_0, \sigma_0), (\mu_0', \sigma_0'))$

   (g) $Loss_C \leftarrow L(Target, Output)$

   (h) **Total loss =** $Loss_{BN} + Loss_G + Loss_C$

   (i) **Update** $I \leftarrow$ backward(Total loss)

---

The *"retro-synthesis"* data generation for a target class $C$ starts with random data of Gaussian distribution $I$ and performing a forward pass on $I$ to obtain intermediate activations and output labels. Then we calculate the aggregated loss that occurs between, stored batch norm statistics and the intermediate activation statistics ($L_{BN}$), the Gaussian loss ($L_G$), and the class loss ($L_C$) between the output of the forward pass and our target output. The $L_2$ loss formulation as in equation 1 is used for $L_{BN}$ and $L_G$ calculation whereas mean squared error is used to compute $L_C$. The calculated loss is then backpropagated till convergence thus generating a batch of retro-synthesis data for a class $C$. The same algorithm is extendable to generate the retro-synthesis data for all classes as well.

$$L((\mu_k, \sigma_k), (\mu_k^{BN}, \sigma_k^{BN})) = \|\mu_k - \mu_k^{BN}\|_2^2 + \|\sigma_k - \sigma_k^{BN}\|_2^2 \tag{1}$$

Where $L$ is the computed loss, $\mu_k$, $\sigma_k$, and $\mu_k^{BN}$, $\sigma_k^{BN}$ are the mean and standard deviation of the $k^{th}$ activation layer and the Batch-Normalization respectively .

By observing the sample visual representation of the retro-synthesis data comparing against the random data depicted in Fig. 1, it is obvious that the retro-synthesis data captures relevant features from the respective image classes in a DNN understandable format. Hence using the retro-synthesis data for the estimation of activation ranges achieves better accuracy as compared to using random data. Also, it outperforms the state-of-the-art data-free quantization methods (Nagel et al., 2019; Cai et al., 2020) with a good accuracy margin when validated on models with and without Batch-Normalization layers. Therefore, the same data generation technique is used in the other two proposed quantization methods (b) and (c) as well.

### 3.2 HYBRID QUANTIZATION

In any quantization method, to map the range of floating-point values to integer values, parameters such as *scale* and *zero point* are needed. These parameters can be calculated either for *per-layer* of

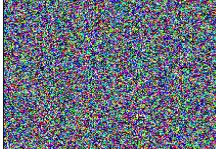 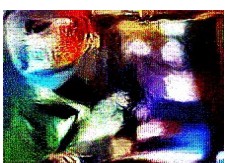

**Random gaussain data**       **Retro-synthesis data**

Figure 1: A sample representation of the random Gaussian data and the Retro-synthesis data using the proposed in Algorithm 1 respectively for a sample class of ResNet-50 model.

the model or *per-channel* in each layer of the model. The former is referred to as *'per-tensor/per-layer quantization'* while the latter is referred to as *'per-channel quantization'*. Per-channel quantization is preferred over per-tensor in many cases because it is capable of handling the scenarios where weight distribution varies widely among different channels in a particular layer. However, the major drawback of this method is, it is not supported by all hardware (Nagel et al., 2019) and also it needs to store scale and zero point parameters for every channel thus creating an additional computational and memory overhead. On the other hand, per-tensor quantization which is more hardware friendly suffers from significant accuracy loss, mainly at layers where the weight distribution varies significantly across the channels of the layer, and the same error will be further propagated down to consecutive layers of the model resulting in increased accuracy degradation. In the majority of the cases, the number of such layers present in a model is very few, for example in the case of MobileNet-V2 only very few depthwise separable layers show significant weight variations across channels which result in huge accuracy loss (Nagel et al., 2019). To compensate such accuracy loss per-channel quantization methods are preferred even though they are not hardware friendly and computationally expensive. Hence, in the proposed *"Hybrid Quantization"* technique we determined the sensitivity of each layer corresponding to both per-channel and per-tensor quantization schemes and observe the loss behavior at different layers of the model. Thereby we identify the layers which are largely sensitive to per-tensor (which has significant loss of accuracy) and then quantize only these layers using the per-channel scheme while quantizing the remaining less sensitive layers with the per-tensor scheme. For the layer sensitivity estimation *KL-divergence (KLD)* is calculated between the outputs of the original FP32 model and the FP32 model wherein the $i$-th layer is quantized using per-tensor and per-channel schemes. The computed layer sensitivity is then compared against a threshold value $(Th)$ in order to determine whether a layer is suitable to be quantized using the per-tensor or per-channel scheme. This process is repeated for all the layers in the model.

The proposed Hybrid Quantization scheme can be utilized for a couple of benefits, one is for accuracy improvement and the other is for inference time optimization. For accuracy improvement, the threshold value has to be set to zero, $Th = 0$. By doing this, a hybrid quantization model with a unique combination of per-channel and per-tensor quantized layers is achieved such that, the accuracy is improved as compared to a fully per-channel quantized model and in some cases also FP32 model. For inference time optimization the threshold value $Th$ is determined heuristically by observing the loss behavior of each layer that aims to generate a model with the hybrid approach, having most of the layers quantized with the per-tensor scheme and the remaining few sensitive layers quantized with the per-channel scheme. In other words, we try to create a hybrid quantized model as close as possible to the fully per-tensor quantized model so that the inference is faster with the constraint of accuracy being similar to the per-channel approach. This resulted in models where per-channel quantization is chosen for the layers which are very sensitive to per-tensor quantization. For instance, in case of *ResNet-18* model, fully per-tensor quantization accuracy is $69.7\%$ and fully per-channel accuracy is $71.48\%$. By performing the sensitivity analysis of each layer, we observe that only the second convolution layer is sensitive to per-tensor quantization because of the huge variation in weight distribution across channels of that layer. Hence, by applying per-channel quantization only to this layer and per-tensor quantization to all the other layers we achieved $10 - 20\%$ reduction in inference time. The proposed method is explained in detail in Algorithm 2. For every layer in the model, we find an auxiliary model $Aux\text{-}model = quantize(M, i, qs)$ where, the step $quantize(M, i, qs)$ quantizes the $i$-th layer of the model $M$ using $qs$ quant_scheme, where $qs$ could be per-channel or per-tensor while keeping all other layers same as the original FP32 weight values. To find the sensitivity of a layer, we find the $KLD$ between the $Aux\text{-}model$ and the original FP32 model outputs. If the sensitivity difference between per-channel and per-tensor is greater than

the threshold value $Th$, we apply per-channel quantization to that layer else we apply per-tensor quantization. The empirical results with this method are detailed in section 4.2.

---

**Algorithm 2** Hybrid Quantization scheme

---

**Input:** Fully trained FP32 model $(M)$ with $n$ layers, retro-synthesis data $(X)$ generated in Part A.
**Output:** Hybrid Quantized Model.

    1. **Init:** quant_scheme$\leftarrow \{PC, PT\}$

    2. **Init:** $M_q \leftarrow M$

    3. **for** $i = 0, 1, \ldots, n$

        (a) $error[PC] \leftarrow 0$ , $error[PT] \leftarrow 0$

        (b) **for** ($qs$ in quant_scheme)

            i. Aux-model$\leftarrow$ quantize($M$,$i$,$qs$).

            ii. Output$\leftarrow M(X)$.

            iii. Aux-output$\leftarrow$ Aux-model$(X)$

            iv. $e \leftarrow$ KLD(Output, Aux-output)

            v. $error[qs] \leftarrow e$

        (c) **if** $error[PT] - error[PC] < Th$
            $M_q \leftarrow$ quantize($M_q, i, PT$)
            **else**
            $M_q \leftarrow$ quantize($M_q, i, PC$)

---

### 3.3 NON-UNIFORM QUANTIZATION

In the uniform quantization method, the first step is to segregate the entire weights range of the given FP32 model into $2^K$ groups of *equal width*, where 'K' is bit precision chosen for the quantization, like K = 8, 6, 4, etc. Since we have a total of $2^K$ bins or steps available for quantization, the weights in each group are assigned to a step or bin. The obvious downside with this approach is, even though, the number of weights present in each group is different, an equal number of steps are assigned to each group. From the example weight distribution plot shown in Fig. 2 it is evident that the number of weights and their range in *'group-m'* is very dense and spread across the entire range, whereas they are very sparse and also concentrated within a very specific range value in *'group-n'*. In the uniform quantization approach since an equal number of steps are assigned to each group, unfortunately, all the widely distributed weights in *'group-m'* are quantized to a single value, whereas the sparse weights present in *'group-n'* are also quantized to a single value. Hence it is not possible to accurately dequantize the weights in *'group-m'*, which leads to accuracy loss. Although a uniform quantization scheme seems to be a simpler approach it is not optimal. A possible scenario is described in Fig. 2, there may exist many such scenarios in real-time models. Also, in cases where the weight distribution has outliers, uniform quantization tends to perform badly as it ends up in assigning too many steps even for groups with very few outlier weights. In such cases, it is reasonable to assign more steps to the groups with more number of weights and fewer steps to the groups with less number of weights. With this analogy, in the proposed Non-Uniform Quantization method, first the entire weights range is divided into three clusters using Interquartile Range (IQR) Outlier Detection Technique, and then assign a variable number of steps for each cluster of weights. Later, the quantization process for the weights present in each cluster is performed similar to the uniform quantization method, by considering the steps allocated for that respective cluster as the total number of steps.

With extensive experiments, it is observed that assigning the number of steps to a group by considering just the number of weights present in the group, while ignoring the range, results in accuracy degradation, since there may be more number of weights in a smaller range and vice versa. Therefore it is preferable to consider both the number of weights and the range of the group for assigning the number of steps for a particular group. The effectiveness of this proposed method is graphically demonstrated for a sample layer of the ResNet-18 model in Fig. 3 in the appendix A.1. By observing the three weight plots it is evident that the quantized weight distribution using the proposed Non-Uniform Quantization method is more identical to FP32 distribution, unlike the uniform

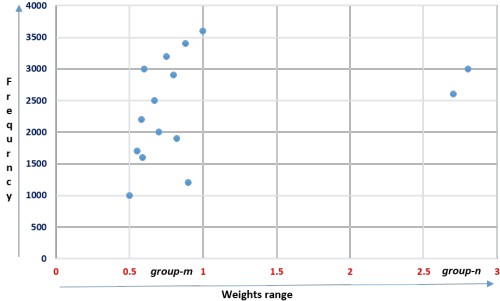

Figure 2: An example weight distribution plot of a random layer in a model with a weight range of $[0 - 3]$ divided into 6 groups of equal width. The ranges from $0.5$ to $1$ and $2.5$ to $3$ are labeled as '*group-m*' and '*group-n*' respectively. For clarity only the weights in these two groups are shown.

quantization method and hence it achieves a better quantized model. Also, it should be noted that the proposed Non-Uniform quantization method is a fully per-tensor based method.

## 4 EXPERIMENTAL RESULTS

### 4.1 RESULTS FOR QUANTIZATION METHOD USING RETRO-SYNTHESIS DATA

Table 1 shows the benefits of quantization using the 'retro-synthesis' data 3.1 against state-of-the-art methods. In the case of models with Batch-Normalization layers, the proposed method achieves $1.5\%$ better accuracy against DFQ and a marginal improvement against ZeroQ. Also, our method outperformed FP32 accuracy in the case of ResNet-18 and ResNet-50. In the case of models without Batch-Normalization layers such as Alexnet and ISONet (Qi et al., 2020) the proposed method outperformed the ZeroQ method by $2 - 3\%$ on the ImageNet dataset.

Table 1: Quantization results using retro-synthesis data for models with and without Batch-Normalization layers on ImageNet dataset with weights and activations quantized to 8-bit (W8A8). BN field indicates whether the respective model has Batch-Normalization layers present in it or not.

| Model | BN | DFQ | ZeroQ | Proposed method | FP32 |
|---|---|---|---|---|---|
| resnet18 | ✓ | 69.7 | 71.42 | **71.48** | 71.47 |
| resnet50 | ✓ | 77.67 | 77.67 | **77.74** | 77.72 |
| mobilenetV2 | ✓ | 71.2 | 72.91 | **72.94** | 73.03 |
| Alexnet | ✗ | - | 55.91 | **56.39** | 56.55 |
| ISONet-18 | ✗ | - | 65.93 | **67.67** | 67.94 |
| ISONet-34 | ✗ | - | 67.60 | **69.91** | 70.45 |
| ISONet-50 | ✗ | - | 67.91 | **70.15** | 70.73 |
| ISONet-101 | ✗ | - | 67.52 | **69.87** | 70.38 |

Table 2 demonstrates the effectiveness of the proposed retro-synthesis data for low-precision (weights quantized to 6-bit and the activations quantized to 8-bit (W6A8)). From the results, it is evident that the proposed method outperformed the ZeroQ method.

Table 2: Results for quantization method using retro-synthesis data with weights quantized to 6-bit and activations quantized to 8-bit (W6A8) on ImageNet dataset

| Model | ZeroQ | proposed method | FP32 |
|---|---|---|---|
| ResNet-18 | 70.76 | **70.91** | 71.47 |
| Resnet-50 | 77.22 | **77.30** | 77.72 |
| MobileNet-V2 | 70.30 | **70.34** | 73.03 |

The efficiency of the proposed quantization method for lower bit precision on the CIFAR-10 dataset for ResNet-20 and ResNet-56 models is depicted in Table 3 below. From the results, it is evident

that the proposed method outperforms the state-of-the-art methods even for lower precision 8, 6, and 4 bit weights with 8 bit activations.

Table 3: Results for quantization method using retro-synthesis data with weights quantized to 8, 6, and 4-bit and activations quantized to 8-bit (W8A8, W6A8, and W4A8) on CIFAR-10 dataset

| Model | W8A8 | | W6A8 | | W4A8 | |
|---|---|---|---|---|---|---|
| | ZeroQ | Proposed method | ZeroQ | Proposed method | ZeroQ | Proposed method |
| ResNet-20 | 93.91 | **93.93** | 93.78 | **93.81** | 90.87 | **90.92** |
| ResNet-56 | 95.27 | **95.44** | 95.20 | **95.34** | 93.09 | **93.13** |

## 4.2 RESULTS FOR HYBRID QUANTIZATION METHOD

Table 4 demonstrates the benefits of the proposed Hybrid Quantization method in two folds, one is for accuracy improvement and the other is for the reduction in inference time. From the results, it is observed that the accuracy is improved for all the models as compared to the per-channel scheme. It should also be observed that the proposed method outperformed FP32 accuracy for ResNet-18 and ResNet-50. Also by applying the per-channel (PC) quantization scheme to very few sensitive layers as shown in "No. of PC layers" column of Table 4, and applying the per-tensor (PT) scheme to remaining layers, the proposed method optimizes inference time by $10 - 20\%$ while maintaining a very minimal accuracy degradation against the fully per-channel scheme.

Table 4: Results on ImageNet dataset with the proposed Hybrid quantization scheme 2 (W8A8), with threshold $Th$ set to 0 for accuracy improvement and set to 0.001 for inference time benefit.

| Model | PC | PT | Hybrid Th=0 | Hybrid Th=0.001 | FP32 | No. of PC layers for Th=0.001 | % time benefit for Th=0.001 |
|---|---|---|---|---|---|---|---|
| Resnet-18 | 71.48 | 69.7 | **71.60** | 71.57 | 71.47 | 1 | 20.79 |
| Resnet-50 | 77.74 | 77.1 | **77.77** | 77.46 | 77.72 | 2 | 17.60 |
| MobilenetV2 | 72.94 | 71.2 | **72.95** | 72.77 | 73.03 | 4 | 8.44 |

## 4.3 RESULTS FOR NON-UNIFORM QUANTIZATION

Since the proposed Non-Uniform Quantization method is a fully per-tensor based method, to quantitatively demonstrate its effect, we choose to compare the models quantized using this method against the fully per-tensor based uniform quantization method. The results with this approach depicted in Table 5, accuracy improvement of **1%** is evident for the ResNet-18 model.

Table 5: Results with Non-Uniform Quantization method (W8A8) on ImageNet dataset

| Model/Method | Uniform quantization | Non-Uniform Quantization | FP32 |
|---|---|---|---|
| ResNet-18 | 69.70 | **70.60** | 71.47 |
| ResNet-50 | 77.1 | **77.30** | 77.72 |

## 5 CONCLUSION AND FUTURE SCOPE

This paper proposes a data independent post training quantization scheme using "retro sysnthesis" data, that does not depend on the Batch-Normalization layer statistics and outperforms the state-of-the-art methods in accuracy. Two futuristic post training quantization methods are also discussed namely "Hybrid Quantization" and "Non-Uniform Quantization" which resulted in better accuracy and inference time as compared to the state-of-the-art methods. These two methods unleashes a lot of scope for future research in similar lines. Also in future more experiments can be done on lower precision quantization such as 6-bit, 4-bit and 2-bit precision using these proposed approaches.

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

## A  APPENDIX

### A.1  NON-UNIFORM QUANTIZATION METHOD

#### A.1.1  CLUSTERING MECHANISM

The IQR of a range of values is defined as the difference between the third and first quartiles $Q_3$, and $Q_1$ respectively. Each quartile is the median of the data calculated as follows. Given, an even $2n$ or odd $2n+1$ number of values, the first quartile $Q_1$ is the median of the $n$ smallest values and the third quartile $Q_3$ is the median of the $n$ largest values. The second quartile $Q2$ is same as the ordinary median. Outliers here are defined as the observations that fall below the range $Q_1 - 1.5IQR$ or above the range $Q_3 + 1.5IQR$. This approach results in grouping the values into three clusters $C_1$, $C_2$, and $C_3$ with ranges $R_1 = [min, Q_1 - 1.5IQR)$, $R_2 = [Q_1 - 1.5IQR, Q_3 + 1.5IQR]$, and $R_3 = (Q_3 + 1.5IQR, max]$ respectively.

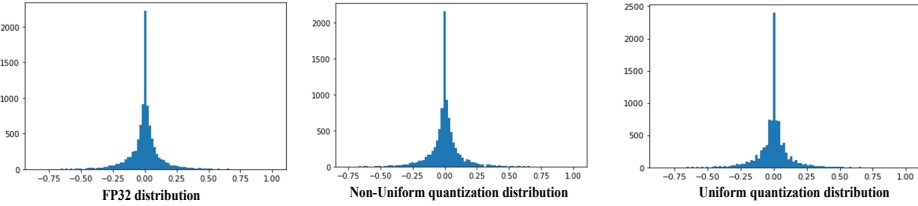

Figure 3: Weight distribution of FP32 model, model quantized using the proposed Non-Uniform Quantization method and uniform quantization method in respective order for a sample layer in ResNet-18 model. The horizontal and vertical axis represents the weights range and frequency respectively.

With extensive experiments it is observed that, assigning the number of steps to a group by considering just the number of weights present in the group, while ignoring the range, results in accuracy degradation, since there may be more number of weights in a smaller range and vice versa. Therefore it is preferable to consider both number of weights and the range of the group for assigning the number of steps for a particular group. With this goal we arrived at the number of steps allocation methodology as explained below in detail.

A.1.2 NUMBER OF STEPS ALLOCATION METHOD FOR EACH GROUP

Suppose $W_i$, and $R_i$ represent the number of weights and the range of $i$-th cluster respectively, then the number of steps allocated $S_i$ for the $i$-th cluster is directly proportional to $R_i$ and $W_i$ as shown in equation 2 below.

$$S_i = C \times (R_i \times W_i) \qquad (2)$$

Thus, the number of steps $S_i$ allocated for $i$-th cluster can be calculated from equation 2 by deriving the proportionality constant $C$ based on the constraint $\Sigma(S_i) = 2^k$, where $k$ is the quantization bit precision chosen. So, using this bin allocation method we assign the number of bins to each cluster. Once the number of steps are allocated for each cluster the quantization is performed on each cluster to obtain the quantized weights.

A.2 SENSITIVITY ANALYSIS FOR PER-TENSOR AND PER-CHANNEL QUANTIZATION SCHEMES

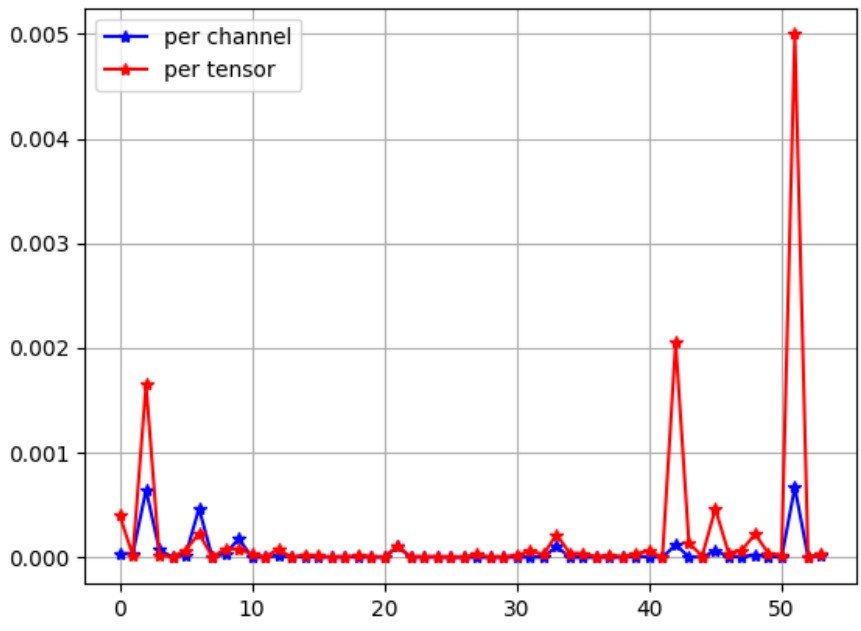

**Sensitivity plot of per-tensor Vs per-channel quantization schemes for MobileNetV2 model**

Figure 4: Sensitivity plot describing the respective layer's sensitivity for per-tensor and per-channel quantization schemes in case of MobileNet-V2 model. The horizontal axis represent the layer number and the vertical axis represents the sensitivity value.

From the sensitivity plot in Fig. 4 it is very clear that only few layers in MobileNetV2 model are very sensitive for per-tensor scheme and other layers are equally sensitive to either of the schemes. Hence we can achieve betetr accuracy by just quantizing those few sensitive layers using per-channel scheme and remaining layers using per-tensor scheme.

A.3 SENSITIVITY ANALYSIS OF GROUND TRUTH DATA, RANDOM DATA AND THE PROPOSED RETRO-SYNTHESIS DATA

From the sensitivity plot inFig. 5, it is evident that there is a clear match between the layer sensitivity index plots of the proposed retro-synthesis data (red-plot) and the ground truth data (green plot) whereas huge deviation is observed in case of random data (blue plot). Hence it can be concluded that the proposed retro-synthesis data generation scheme can generate data with similar characteristics as that of ground truth data and is more effective as compared to random data.

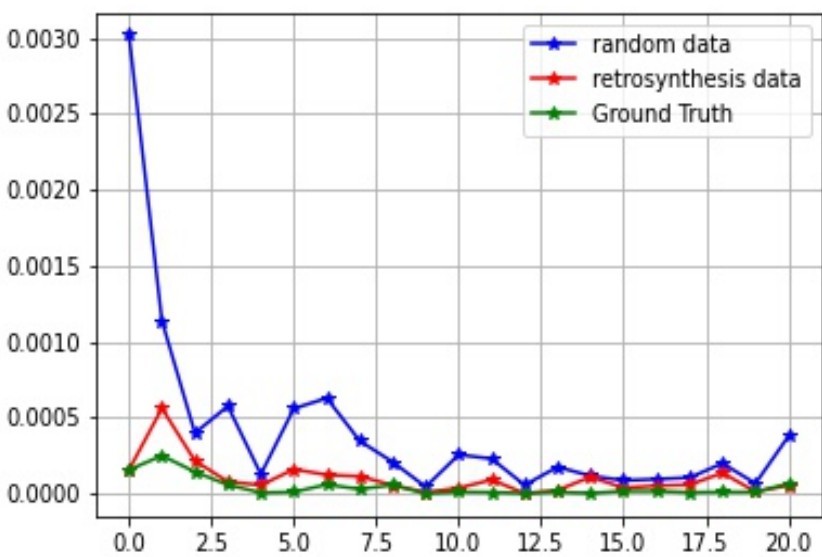

**Sensitivity plot comparing Ground truth data Vs random data Vs retrosynthesis data**

Figure 5: Sensitivity plot describing the respective layer's sensitivity for original ground truth dataset, random data and the proposed retro-synthesis data for ResNet-18 model quantized using per-channel scheme. The horizontal axis represent the layer number and the vertical axis represents the sensitivity value.

