# OpenReview forum: "Hybrid and Non-Uniform DNN quantization methods using Retro Synthesis data for efficient inference"
_ICLR.cc/2021/Conference — Reject_

### Official Review · AnonReviewer4 · 2020-10-25
**Official Blind Review #4**

**Rating:** 4
**Confidence:** 5

**Review:**

The paper proposes a data-independent post-training quantization scheme by generating a faux dataset without depending on the BN layer statistics of the FP32 model. The authors also introduce two variants of post-training quantization methods, hybrid quantization and non-uniform quantization methods.

Strengths of the paper:
- The proposed quantization method is practically applicable to privacy-constraint applications because no training data are required and the method works for any model.
- The proposed Hybrid Quantization scheme addresses a couple of benefits, one is for improving accuracy and the other is for faster inference time. Both of them are important metrics for practical applicability.
- The paper is well written and easy to follow.

Weaknesses of the paper:
- Novelty: The novelty is moderate but not strong enough. Specifically, (1) a zero-shot quantization framework to generate a synthetic dataset has been studied in the previous literature ZeroQ (using the distilled data engineered to match the statistics of BN layers to perform post-training quantization). But this paper employs an alternative faux dataset, called by the Retro-Synthesis Data, that does not depend on the BN layer statistics of the original data, which should be appreciated. Moreover, most of DNN models employ the BN layer in their construction, what is the merit in that case? (2) The same layer sensitivity metric as ZeroQ, the KLD between the original model and the quantized model has been used in the proposed Hybrid Quantization scheme. The main idea behind Hybrid Quantization using the KLD to determine whether a layer is suitable to be quantized using the per-tensor or per-channel scheme, seems to be same. Overall, the contribution is a bit incremental and seems not novel to me compared with ZeroQ and DFQ.
- Evaluation: Further experiments and more ablation studies should be done. The provided experimental results are too weak to support the strength of using the retro-synthesis data.

Detailed comments:

(1) Algorithm 1 to generate retro-synthesis data seems to have a dependency on the target class(C). If there are many classes to be classified, is the time complexity also increasing? The paper states that generating the retro-synthesis data takes 10~12 sec. Is this for CIFAR-10 or ImageNet dataset? The proposed scheme seems to be more time-consuming than ZeroQ's 3 sec for generating the distilled data. In the ZeroQ paper, the end-to-end quantization of ResNet50 on ImageNet data takes almost 30 sec, which takes 3 sec to generate the synthetic data, 12 sec to estimate the sensitivity for all layers, and 14 sec to perform Pareto Frontier optimization etc. Could you provide the detailed timing breakdown on which experimental setting?

(2) Computing the KLD in Algorithm 2 has more timing O/H than ZeroQ since it is executed twice for both per-tensor and per-channel quantization schemes per layer. Then does it take time twice more compared with ZeroQ? Could you provide the detailed timing breakdown or time O/H evaluation on your Hybrid Quantization configuration? Is the time O/H ignorable compared to the benefit of faster inference time by exploiting the hybrid quantization scheme? Approaches to achieve the goal of optimizing the inference time cause a conflict of interest.

(3) A threshold value Th is sub-optimal and determined heuristically to decide either per-channel or per-tensor quantization. Is there any way to find an optimal value? The paper uses Th=0.001, but there is little improvement over Th=0. Could you explain why this happens? Further ablation study on different Th values should be provided. In the 8-bits setting, there seems no significant difference b/w per-channel and per-layer schemes due to its perturbation effect. More explanation on the result is required.

(4) Results (in Table 3) of comparing the proposed method to ZeroQ with W8A8, W6A8, and W4A8 on ResNet model with CIFAR-10 are too weak to support the strength of using the retro-synthesis data. There seems to be a bit improvement over ZeroQ while the proposed scheme takes more time(10~12 sec for the retro-synthesis data) to generate a synthetic dataset compared with ZeroQ(3 sec for the distilled data).

(5) Results on ImageNet data (in Table 4 and 5) have a weak contribution if the bitwidth used in the experiment is 8 bit(You should describe a precision setting in the result tables). More aggressive setting of bitwidths, say 6 or 4 bits, should be provided comparing with the SOTA post-training quantization schemes.

(6) The paper also presents another variant of using the retro-synthesis data, a non-uniform quantization scheme. Further evaluation (in Table 5) should be done to support its superiority by comparing it with the SOTA non-uniform quantization schemes, e.g. PoT, APoT, DDQ, etc. Especially, the non-uniform quantization gets more important in the weaker representation range, say low-precision quantization schemes below 8 bits. Future more experiments should be provided.

---

> ### Author Response · Authors · 2020-11-18
> **Response to AnonReviewer4**
>
> Thanks to the reviewer for the detailed comment and feedback
>
> Before Addressing each of the comments, we would like to provide more details about the proposed methodology
>
> .1) Keeping in mind the concerns about timings for quantizing the model, we would like to clarify that all the three proposed methods namely retro-synthesis data, hybrid quantization and non-uniform quantization are offline quantization schemes and hence the data generation process, sensitivity analysis etc. are done before deployment of the model and not during runtime.
>
> .2)The proposed hybrid quantization method is not an extension/alternative of ZeroQ’s mixed precision method. Previous works have shown [1], [2] that a fully per-tensor quantization leads to accuracy loss whereas a fully per-channel quantization is not hardware friendly, because of a different zero-point and scale values are associated per each channel. Hence we proposed a hybrid quantization method that uses a combination of per-tensor and per-channel schemes for different layers for achieving accuracy and performance improvement. Though we have used a similar method as that of ZeroQ for sensitivity calculation our goal is very different. So comparing the proposed hybrid-quantization scheme and the ZeroQ’s mixed precision scheme would be of less relevance.

---

> > ### Author Response · Authors · 2020-11-18
> > **Response to AnonReviewer4 [continue]**
> >
> > Following are our responses to respective comments:
> >
> > .1) Here in the retrosynthesis data generation method we have introduced an extra class loss component. The calculation of class loss (step e in algo1) involves the number of classes, so the time to generate data will depend on the number of classes and hence the retro-synthesis data generation takes more time than distilled data. The 10-12 secs of time  taken is to generate the data for the resnet50 model trained on Imagenet dataset. The end to end quantization of resnet50 using hybrid quantization method takes ~20secs, 12sec to generate data and 7-8secs for sensitivity estimation for both per-channel and per-tensor quantization on a GTX1080Ti system (more explanation for sensitivity estimation timing in next point)
> >
> > .2) As seen in eq2 of zeroq [3], it is calculating the KLD thrice for sensitivity estimation for  k=[2,4,8]. However in our method we are doing only twice, for per-tensor and per-channel, hence there is no timing O/H as compared to ZeroQ. Regarding conflict of interest for optimizing the inference time, determining the configuration of each layer for per-channel or per-tensor is done beforehand and not during runtime, hence there is no timing O/H created during inference due to sensitivity calculation.
> >
> > .3) For our experiments we have determined the value of Th from the sensitivity plot (fig 4 of the updated rebuttal version) from which it is becoming conclusive. So, to answer: Is there any way to find an optimal value? We have not formed any optimization problem as such to solve for Th. In 8-bit settings a significant difference can be observed between per-channel and per-channel accuracy as mentioned in Table-4.
> >
> > .4) We agree that the improvement shown in Table-3 for the Cifar10 dataset is not very significant whereas significant improvement can be observed from results of Table-1. In our humble opinion it is harder to achieve even a marginal improvement over an strong baseline of ZeroQ, which is already close to fp32 accuracy.
> >
> > The mentioned time complexity of 10-12sec for retro-synthesis data generation is for ImageNet dataset, whereas for Cifar-10 dataset it is only 2-3sec  as there are just 10 classes.
> >
> > .5) & 6) In the current scope of this paper we have analyzed the proposed hybrid quantization and non-uniform quantization schemes only for 8-bit precision. We will definitely perform lower precision analysis in the future work. Table-4 and Table-5 are for 8bit precision. We have added the detail in the revised version. Thanks to the reviewer for pointing it out
> >
> > References
> >
> > [1] Raghuraman Krishnamoorthi. Quantizing deep convolutional networks for efficient inference: A whitepaper. arXiv preprint arXiv:1806.08342, 2018
> >
> > [2] Markus Nagel, Mart van Baalen, Tijmen Blankevoort, and Max Welling. Data-free quantization through weight equalization and bias correction. In Proceedings of the IEEE International Conference on Computer Vision, pp. 1325–1334, 2019
> >
> > [3] Yaohui Cai, Zhewei Yao, Zhen Dong, Amir Gholami, Michael W Mahoney, and Kurt Keutzer. Zeroq: A novel zero shot quantization framework. In Proceedings of the IEEE/CVF Conference on Computer Vision and Pattern Recognition, pp. 13169–13178, 2020.

---

### Official Review · AnonReviewer2 · 2020-10-27
**Post-training quantization without dataset access to preserve privacy**

**Rating:** 6
**Confidence:** 5

**Review:**

This work uses post-training quantization without access to training data for privacy concerns. Instead, useful statistics are estimated using a retro-synthesis data obtained from the FP baseline. I have a few comments, some concerns and some suggestions I think can be used to improve this work.

Overview of prior work:
Most competing approaches are included and representative prior arts are mentioned which is good. In the descussion about post-training quantization methods, the authors conclude that prior arts observe accuracy degradations. There have been works on post-training quantization that theoretically predict such degradation and overcome it by increasing precision as needed using the concept of noise gains [1]. I think the authors should contrast such work with theirs. It is my impression that noise gains in [1] can be used to improve the presented method.

Comments about the method:
Retro-synthesis data generation:
The retro-synthesis data is used to determine activation ranges which are useful for quantization. This is a clever method. I wonder if that is really necessary though. Can't we follow an analysis similar to [2] in order to predict activation statistics from weight statistics (which are available)?

Hybrid quantization:
It seems the concept of per-tensor quantization has already been studied in [3] and also use the concept of noise gains as above to analytically determine the required number of bits for each tensor. I think this method can be useful to improve/validate the proposed hybrid quantization technique.

Non-uniform quantization:
I have two issues: (1) why not simply use the Lloyd-Max algorithm (which optimizes non-uniform quantization), (2) it is unclear how activations are quantized.

The experimental results look good.

[1] Sakr, C., Kim, Y., & Shanbhag, N. Analytical guarantees on numerical precision of deep neural networks. In International Conference on Machine Learning, ICML 2017.

[2] He, K., Zhang, X., Ren, S., & Sun, J. Delving deep into rectifiers: Surpassing human-level performance on imagenet classification. In Proceedings of the IEEE international conference on computer vision, CVPR 2015.

[3] Sakr, C., & Shanbhag, N. Per-tensor fixed-point quantization of the back-propagation algorithm. In 7th International Conference on Learning Representations, ICLR 2019.

Post Rebuttal Comments:

I thank the authors for their feedback. I have no modification to make to my original review.

---

> ### Author Response · Authors · 2020-11-17
> **Response to  AnonReviewer2**
>
> Thanks to the reviewer for the valuable feedback and suggestions. Please find our responses below for respective queries
>
> Comments about the method [Author’s Response]: Thank you for introducing the reference [1]. We will definitely try to contrast our work with [1] in the final submission version else will add it as a part of our future work scope.
>
>
> We agree that there are methods like [2] to find correlation between the weight distribution statistics and activation statistics. However as mentioned in Figure-2 of (Nagel et al. 2019) incase of models like MobileNetV2, where we observe huge variation in weight ranges across the channels of some layers, the correlation may not hold good. In such cases, we need a dataset to accurately estimate the activation ranges. As it is not possible to get access to the dataset always, the proposed retro-synthesis scheme is very much needed.
>
> Hybrid quantization [Author's response]:The proposed Hybrid quantization scheme is a novel method that analyzes a combination of per-channel and per-tensor approaches for achieving better accuracy and performance. In this scope of the paper we have analyzed the proposed hybrid quantization scheme for 8-bit precision. As per your suggestion we try to apply the method in [3] to analyze our hybrid approach for lower bit precision (6, 4 and 2 bits) as well as a future scope of work. Thank you for the suggestion.
>
> Non-uniform quantization [Author’s response:]:
>
> (1) The two obvious choices that strike the thought are K-means and Lloyd–Max quantizer. However, one of the key features of CNN is the large numbers of weights and these weights ranges are not very wide and these have similar values. Hence For K-means it will take a lot of time to reconstruct a codebook as distance calculation for all weights is not efficient. On the other hand, Lloyd-max quantizer performance is also affected by the number of data points when performing an integrated calculation in the actual computing environment. Hence we have chosen the proposed method in contrast to the above methods.
>
> (2) Only the weights are quantized using the proposed non-uniform approach, the activations are quantized using asymmetric uniform quantization.
>
> References:
>
> Markus Nagel, Mart van Baalen, Tijmen Blankevoort, and Max Welling. Data-free quantization through weight equalization and bias correction. In Proceedings of the IEEE International Conference on Computer Vision, pp. 1325–1334, 2019
>
> Please let us know for any further queries.

---

### Official Review · AnonReviewer3 · 2020-10-28
**OK ideas that come across as incremental; good numbers but questionable science.**

**Rating:** 4
**Confidence:** 4

**Review:**

This paper considers the problem of data-free post-training quantization of classfication networks. It proposes three extensions of an existing framework ZeroQ (Cai et al., 2020): (1). in order to generate distilled data for network sensitivity analysis, the "Retro Synthesis" method is proposed to turn a random image into a one that represents a desired class label without relying on batch norm statistics like in ZeroQ; (2). a hybrid quantization strategy is proposed to optionally provide finer-grained per-channel quantization instead of the typical per-layer quantization; (3). a non-uniform quantization grid is proposed to better represent quantized weights, instead of uniform quantization as in ZeroQ.  Empirical evaluation demonstrate the effectiveness of the proposed approach.

==========================================================

Pros:
1. The proposed "Retro Synthesis" approach broadens the scope of ZeroQ to a wider range of neural network architectures by lifting the requirements of having batch normalization layers, and thus may significantly extend the practical applicability of data-free post-training quantization.
2. The proposed approach seems to run very fast (but on the same order of magnitude as ZeroQ).


==========================================================

Cons:
1. The significance of the contribution appears limited in scope, as the three proposed methods (especially the latter two, "hybrid quantization" and non-uniform quantization) read more like incremental modifications to components of the existing ZeroQ framework. As far as I understood, the overall approach of this paper follows the ZeroQ framework, e.g., uses the generated distilled data in the same way, also computes the effect of quantization based on KL divergence, and allocates per-layer/channel bit precision using the same Pareto frontier approach. The proposed "Retro Synthesis" is the main innovation, but is rather poorly explained (what's the motivation for having three components L_BN, L_G, L_C, to the total loss, and are they really weighted equally in the experiments?) and lacks insight into why it works.  And it is questionable whether the remaining two proposals count as contributions: "hybrid quantization" is a simple heuristic for deciding when to use per-channel vs per-layer quantization, and non-uniform quantization is a slightly more sophisticated quantization grid (simply divides weights into quartiles and does uniform quantization in each quartile). I have no issue with a simple method if it's well motivated and works well, but:
2. Moreover, the experiments are plagued with a lack of careful analysis of the proposed methods and details for reproducability. Since the three proposed improvements focus on different aspects of ZeroQ, ablations that analyze the effect of each in isolation  (i.e., baseline ZeroQ + proposed hybrid quantization, baseline ZeroQ + proposed non-uniform quantization) are crucial for evaluating their contributions. Details such as how the pre-trained models were obtained (were they trained from scratch, or pre-trained and obtained from the original ZeroQ authors) are not provided, which can significantly impact the empirical results and how they can be understood.  The issue is not specific to this paper -- the field of neural network compression can benefit from better reproducability and more robust evaluation, and particularly methods that operate post-training can evaluate on a shared repository of pre-trained models and report accuracy loss.

==========================================================

Comments & Questions:
1. Since the paper argues that "it is possible to generate image data with similar statistics" based on class scores (under the proposed "Retro Synthesis" approach), it makes sense to validate this is indeed the case qualitatively by comparing against the alternative "ground truth" approach based on matching batch norm statistics, e.g. by comparing the resulting model layer sensivity on distilled data produced by these two approaches, like in Figure 2 of the ZeroQ paper.
2. What's the Gaussian loss (LG) component of the main loss function (eq (1))? Similarly, step (e) of algorithm 1 is unclear -- what is \mu_0' and \sigma_0'?
3. Maybe I'm missing something, but wouldn't something obvious like k-means perform just as well (if not better) compared to the proposed non-uniform quantization approach based on uniformly quantizing within quartiles?

---

> ### Author Response · Authors · 2020-11-18
> **Response to AnonReviewer3**
>
> Thanks to the reviewer for the valuable feedback and suggestions. Please find our responses below for respective queries.
>
> [Query:] As far as I understood, the overall approach of this paper follows the ZeroQ framework, e.g., uses the generated distilled data in the same way, also computes the effect of quantization based on KL divergence, and allocates per-layer/channel bit precision using the same Pareto frontier approach
>
> [Author’s Response:] The proposed Hybrid Quantization uses the KLD method for sensitivity estimation similar to ZeroQ, but with altogether a different goal to determine whether a layer can be quantized using a per-tensor or per-channel quantization scheme. Whereas ZeroQ uses it for  accurately estimating the required bit-width in their mixed precision scheme. Apart from accuracy improvement Hybrid Quantization also addresses a very practical aspect of DNN model deployment on edge/mobile devices which is the inference time. We have shown an inference time improvement of ~10-20% by benchmarking the models on Samsung Galaxy S20 mobile platform with Qualcomm Hexagon DSP hardware. This aspect of performance improvement is not discussed by ZeroQ. Also, as per our knowledge, a combination of per-channel and per-tensor quantization has not been explored for accuracy or performance improvement in existing literature
>
> [Query:]  The proposed "Retro Synthesis" is the main innovation, but is rather poorly explained (what's the motivation for having three components L_BN, L_G, L_C, to the total loss, and are they really weighted equally in the experiments?) and lacks insight into why it works
>
> [Author’s Response:] How effectively the proposed "Retro-Synthesis Data" can represent the desired data classes as compared to random data is depicted in Fig.1 of the paper. Also, When the actual training/testing data is not available, instead of using the random data the benefits of resorting on the proposed "Retro-Synthesis Data" is shown in Fig.5 of the uploaded rebuttal version. From the sensitivity plot in Fig. 5 it is evident that there is a clear match between the layer sensitivity index plots of the proposed retro-synthesis data (red-plot) and the ground truth data (green plot) whereas huge deviation is observed in case of random data (blue plot). Hence it can be concluded that the proposed retro-synthesis data generation scheme can generate data with similar characteristics as that of ground truth data and is more effective as compared to random data
>
> L_BN: the aggregated loss between, stored actual batch norm statistics of the model and the intermediate activation statistics computed using generated retro-synthesis data for every iteration.
> L_G: Loss between generated retro-synthesis data distribution and the gaussian distribution. This loss component makes sure that the generated retro-synthesis data is normally distributed.
> L_C: Loss between the softmax output of the forward pass and target vector for class C. This loss is introduced to ensure the generated retro-synthesis data for an image class have the same statistics matching to the corresponding class of the original or actual dataset.
> Yes, All the three losses are equally weighted
>
> [Query] : Since the three proposed improvements focus on different aspects of ZeroQ, ablations that analyze the effect of each in isolation (i.e., baseline ZeroQ + proposed hybrid quantization, baseline ZeroQ + proposed non-uniform quantization) are crucial for evaluating their contributions.
>
> [Author’s Response] : Table-1 of the paper compares baseline ZeroQ method with our proposed retro-synthesis data baseline method. From the results of Table-1, it is evident that our baseline method outperforms ZeroQ baseline results. Hence it is clear that the retro-synthesis data generation method is more accurate than the distilled data in ZeroQ. So, for the subsequent experiments to showcase the effectiveness of the proposed Hybrid Quantization and non-uniform quantization methods we chose to compare them against the proposed baseline method rather than ZeroQ baseline method. However we can add the comparison results of ZeroQ baseline+hybrid method and ZeroQ baseline + non uniform in the final version of the paper.
>
> [Query]: Details such as how the pre-trained models were obtained (were they trained from scratch, or pre-trained and obtained from the original ZeroQ authors) are not provided, which can significantly impact the empirical results and how they can be understood.
>
> [Author’s Response] We have used pre-trained models from  PytorchCV (https://pypi.org/project/pytorchcv/) which are the same as what ZeroQ has used. We will add this detail in the final version of the paper.

---

> > ### Author Response · Authors · 2020-11-18
> > **Response to AnonReviewer3 [continue]**
> >
> > Comment and Questions:
> >
> > 1) Thanks for the valuable suggestion. The benefits of resorting on the proposed "Retro-Synthesis Data" is shown in Fig.5 of the uploaded rebuttal version. From the sensitivity plot in Fig. 5 it is evident that there is a clear match between the layer sensitivity index plots of the proposed retro-synthesis data (red-plot) and the ground truth data (green plot) whereas huge deviation is observed in case of random data (blue plot). Hence it can be concluded that the proposed retro-synthesis data generation scheme can generate data with similar characteristics as that of ground truth data and is more effective as compared to random data.
> >
> > 2) The Gaussian Loss (L_G) component makes sure that the generated data is normally distributed. $\mu_0'$ and $\sigma_0'$ are the mean and standard deviation of the current batch of input images respectively. Gaussian loss (L_G) tries to keep $\mu_0'$` and $\sigma_0'$ close to 0 and 1 respectively to assure that the generated retro-synthesis data is normally distributed.
> >
> > 3) Non uniform quantization is proposed for weight quantization because it optimizes to reduce the quantization error. The obvious choice that comes into picture for non-uniform quantization is K-means. However, one of the key features of CNN is the large numbers of weights and these weights ranges are not very wide and also these have similar values. Hence for K-means it will take a lot of time to reconstruct a codebook as distance calculation for all weights is not efficient. Hence we have chosen the proposed non-uniform quantization method in contrast to the K-means.
> >
> > Please let us know for any further queries.

---

### Official Review · AnonReviewer1 · 2020-10-29
**A nicely written paper, but it might need more novelty**

**Rating:** 4
**Confidence:** 4

**Review:**

This paper proposed three different techniques to improve the quality of the post-training quantization (PT) results. The main claim is about Retro-Synthesis Data, which allows the calibration of the quantization parameters without the training data. There are two additional techniques, Hybrid Quantization and Non-Uniform Quantization, to increase the accuracy of PT.

Although the overall paper is nicely written, the contents might need more novelty for being accepted in ICLR. Here are a few concerns related to it.

- The "Retro-Synthesis Data" scheme sounds interesting, but not sure how much practical impact it might bring in. The authors assumed the model deployment scenario where the full-precision model is given without any training data. But it does not necessarily mean that the unlabeled data is also not available; in most cases, test data (without labels) from the deployed application is available for tuning of parameters (E.g., Sec 4 of [Sun et al., NeurIPS19]). In such a case, it is not clear how much benefit we can expect from "synthesizing" the test data for tuning quantization parameters.

- Even if we believe that synthesizing the test data is necessary, it is not clear why "Retro-Synthesis Data" is well suited for the post-training quantization. Although the approach is interesting, how much representative the data created by the proposed method is for being used for the parameter tuning? It would be highly desirable to explain in the derivation of "Retro-Synthesis Data" how it can provide information particularly useful for better post-training quantization.

- The strong empirical evidence might say more than just a detailed explanation. But unfortunately, the performance gain in the experimental results seems to be marginal. The authors claim that the proposed method is good for the models without batchnorm, but it does not sound very convincing given the fact that batchnorm is extremely popular these days...

- The additional techniques, Hybrid Quantization and Non-Uniform Quantization, seem to be incremental from the previous work about quantization-sensitivity and outlier aware controls of neural networks (such as precision selection and channel splitting). Also, the authors claimed that there is 10~20% performance gain from Hybrid Quantization, but without much justification on how such hardware performance is measured/estimated.

To sum, this is a well-written paper but it might need more novelty for being accepted in ICLR.

---

> ### Author Response · Authors · 2020-11-18
> **Response for AnonReviewer1**
>
> Thanks to the reviewer for the valuable feedback and suggestions. Please find our responses below for respective queries
>
> • We agree with you that, synthesizing the test data may not be needed in general, but when we consider scenarios where the hardware is customized for quantized models, such as cloud based deep learning inference providers or smartphone providers there is obvious necessity to provide a generic quantization service (FP32 models can be converted to lower precision), without needing the data or receiving data from their customers to fine-tune the models. Else they become strictly dependent on the model developers to share their data or should expect the model developers to do quantization. Both of these cases add an additional overhead to the entire process. Even for model developers having the capability to quantize their own models without the need to generate actual data results in quicker validation of the models and saves significant time.
> Also, in case of data privacy applications which involve customer sensitive information such as medical records, credit card details, security numbers etc. since it is very difficult to get access to necessary data from the customers or may have access to very less amount of data, the proposed method is very helpful to generate the data in required quantity for accurately fine tuning the parameters.
>
> • How effectively the proposed "Retro-Synthesis Data" can represent the desired data classes as compared to random data is depicted in Fig.1 of the paper. Also, When the actual training/testing data is not available, instead of using the random data the benefits of resorting on the proposed "Retro-Synthesis Data" is shown in Fig.5 of the uploaded rebuttal version. From the sensitivity plot in Fig. 5 it is evident that there is a clear match between the layer sensitivity index plots of the proposed retro-synthesis data (red-plot) and the ground truth data (green plot) whereas huge deviation is observed in case of random data (blue plot). Hence it can be concluded that the proposed retro-synthesis data generation scheme can generate data with similar characteristics as that of ground truth data and is more effective as compared to random data.
>
> • 1) Despite the fact that the proposed approach achieves marginal improvement over state-of-the-art methods in case of models with batchnorm layers, it should be noted that the mentioned state-of-the-art methods achieve accuracy close to FP32 accuracy and at that level it becomes exponentially harder to achieve  accuracy gains. Hence in our humble opinion achieving those margins is a significant improvement as compared to state-of-the-art methods.  Also the effectiveness of the proposed method is evident in case of ResNet-18 and ResNet-50 models where it outperformed FP32 accuracy.
> 2) The idea behind designing the approach independent of the batchnorm layers is to avoid any limitation that refrains anyone from using the proposed method. As mentioned in [1], the major challenge with the commonly used normalization layers in modern networks require certain statistical independence assumptions to hold and large enough batch size or channel number for precise estimation of such statistics. This drawback significantly limits their applications to robust learning [2], contrastive learning [3], implicit models [4], object detection etc.
> The ISONETs [1] which does not have batchnorm layers may stand as inspiration for many such model designs in future, and as stated in our paper, the existing state-of-the-art data free quantization methods fail to quantize this model.
>
> • 1) The proposed Hybrid Quantization technique is an unique approach encapsulating both per-tensor & per-channel schemes in it by judiciously choosing the respective scheme for each layer based on the computed sensitivity for accuracy loss or performance gain. Whereas the proposed Non-uniform quantization method is a per-tensor based scheme with varied bins/steps  allocated for respective weight ranges in each layer to achieve better accuracy as compared to per-tensor uniform quantization method. Hence in our opinion the proposed two methods follow different approaches unlike the mentioned precision selection and channel splitting.
> 2) The mentioned performance improvement of 10-20% is measured by quantizing and benchmarking the respective models using fully per-channel, fully per-tensor and the proposed Hybrid Quantization approaches on Samsung Galaxy S20 mobile platform having Qualcomm Hexagon DSP hardware.

---

> > ### Author Response · Authors · 2020-11-18
> > **Response for AnonReviewer1 [Continue]**
> >
> > references
> >
> > [1] Haozhi Qi, Chong You, Xiaolong Wang, Yi Ma, and Jitendra Malik. Deep isometric learning for visual recognition. arXiv preprint arXiv:2006.16992, 2020
> >
> > [2] Sun, Y., Wang, X., Liu, Z., Miller, J., Efros, A. A., and Hardt, M. Test-time training for out-of-distribution generalization. arXiv, 2019
> >
> > [3] Chen, T., Kornblith, S., Norouzi, M., and Hinton, G. A simple framework for contrastive learning of visual representations. arXiv, 2020
> >
> > [4] Bai, S., Koltun, V., and Kolter, J. Z. Multiscale deep equilibrium models. arXiv, 2020

---

### Author Response · Authors · 2020-11-17
**Updated version of the paper**

Thanks to All the reviewers for reviewing our work and providing constructive feedback. As per the reviewers' comments, we have updated our manuscript with the following details in the Appendix section to provide better clarity about the proposed method and to demonstrate its effectiveness in comparison to state-of-the-art methods.

• Figure-4 with the sensitivity plot describing the respective layer's sensitivity for per-tensor and per-channel quantization schemes in case of MobileNet-V2 model.
The plot clearly shows that only a few layers in the MobileNetV2 model are very sensitive to the per-tensor scheme and other layers are equally sensitive to either of the schemes. Hence we can achieve better accuracy by just quantizing those few sensitive layers using per-channel scheme and the remaining layers using per-tensor scheme.

• Figure-5 with sensitivity plot describing the respective layer's sensitivity for the original ground-truth dataset, random data, and the proposed retro-synthesis data for ResNet-18 model quantized using the per-channel scheme.
From the sensitivity plot, it is evident that there is a clear match between the layer sensitivity index plots of the proposed retro-synthesis data (red-plot) and the ground truth data (green plot) whereas a huge deviation is observed in the case of random data (blue plot). Hence it can be concluded that the proposed retro-synthesis data generation scheme can generate data with similar characteristics as that of ground truth data and is more effective as compared to random data.

---

### Decision · Program_Chairs · 2021-01-07
**Final Decision**

**Decision:**

Reject

**Comment:**

Four knowledgeable referees reviewed this paper; one reviewer (weakly) supports accept and other three indicate reject. Even with the rebuttal, all negative reviewers have concerns on the limited novelty and marginal performance improvement, and agree that the paper is not well qualified for the high standard of ICLR.